# Israeli Jewish Attitudes toward Core Religious Beliefs in God, the Election of Israel, Eschatology, and the Temple Mount—Statistical Analysis

**Motti Inbari [1,*] and Kirill M. Bumin [2,*]**

[1]  Department of Philosophy and Religion, The University of North Carolina Pembroke, Pembroke, NC 28372, USA
[2]  Metropolitan College, Boston University, Boston, MA 02215, USA
*   Correspondence: inbari@uncp.edu (M.I.); kmbumin@gmail.com (K.M.B.)

**Abstract:** In this article, we aim to gauge the perspectives of Israeli Jews on core Jewish beliefs in God, the ideas of the Election of Israel, the afterlife, the advent of the messiah, and the significance of the Temple Mount at the End of Days. We conducted a survey among a representative sample of 1204 Israeli–Jewish respondents. The survey was administered in Hebrew and fielded between 27 March and 18 April 2023. This study shows that among the Israeli public, there is a so-called secular–religious dichotomy, at least to some extent. We were able to confirm that about 50% of the sample prays often, believes that Judaism is the only true religion, and identifies as traditional or Orthodox. We also analyze a typology of secular Israelis, including traditionalist seculars, spiritual seculars, and atheist or agnostic seculars. This study further shows that there are important generational differences in Israeli society when it comes to questions of faith. The youngest Israelis comprise the most religious age cohort, while the older generations are the least religious. In the survey, we asked multiple questions on the opinions toward visiting, praying, and constructing a synagogue or a Temple on the Temple Mount.

**Keywords:** Israeli Jews; beliefs; survey data; chosen people; promised land; afterlife; world to come; messiah; temple mount; Jewish eschatology





## 1. Introduction

In this article, we aim to gauge the perspectives of Israeli Jews on core Jewish beliefs in God, the ideas of the Election of Israel, and eschatology: the belief in an afterlife, the advent of the messiah, and the significance of the Temple Mount at the End of Days. To delve into these topics, we conducted a survey among a representative sample of 1204 Israeli–Jewish respondents.

The belief in God and his providence and the Biblical idea of the Election of Israel are historically some of the most fundamental Jewish ideologies. The beliefs in the afterlife, heaven, and hell, and the coming of the messiah, are also essential Jewish ideas (although some rabbinical authorities understood them metaphorically). Jewish eschatological beliefs are part of a Jewish theodicy—an ideological attempt to reconcile the existence of an all-powerful, all-good God with evil that exists in the world. In the broad sense, theodicy is an ideological attempt to explain anomic events and make sense of evil (Kessler 2006, p. 182). Eschatological theodicies attempt to make the experience of what might be pointless pain and suffering meaningful by promising that, someday, things will be made right and justice will triumph. Eschatological beliefs are already mentioned in the prophetic literature of the Bible, and during Second Temple Judaism, different eschatological views prevailed among the different sects (Hengel 1989; Horsley and Hanson 1986, 1985; Ben-Shalom 1993).

Jewish eschatology includes two types: this-worldly and otherworldly. This-worldly eschatology centered on what happens to us when we die, and, thus, ideas of life after death,

the world to come, heaven and hell, reincarnation, and resurrection of the dead have become prominent since the Second Temple period (Sievers 2010). Beyond personal salvation, Judaism also developed ideas of national redemption. Expectations for a messianic figure who would deliver Jews from oppression crystallized during the Second Temple period and throughout the two thousand years of Jewish exile. The expectation for the coming of the messiah and the miraculous End Time that would put an end to Jewish suffering in exile had been popular among Jews. Over time, several messianic figures became prevalent among the Jewish masses (Scholem 1976; Heilman and Friedman 2010). To what extent are these beliefs still relevant among the Israeli Jews today?

Jewish messianic beliefs center on the notion that, one day, all Jews would return to their homeland and restore their institutions as they existed in antiquity, including the restoration of the Davidic monarchy, the rebuilding of the Temple on the Temple Mount, and the restoration of the Sanhedrin (Supreme Court). During medieval times, Jewish scholars debated whether these restorations would be miraculous or part of human effort (Hartman 1985, pp. 179–85; Schwartz 1997). With the rise of the Zionist movement and, later, the establishment of the State of Israel, many contemplated the question of whether secular Zionism is a temporal fulfilling messianic expectation (Ravitzky 1993). In this article, we will seek to examine cases in which these religious beliefs are related to the modern Israelis living in Israel, the highly industrialized "start-up nation". (Senor and Singer 2011).

One of the tensions of Israeli culture surrounds its Jewish characteristics. Israel's Declaration of Independence stated that it is a *Medinah Yehudit* (a Jewish state), and a lot of ink has been spilled in order to explain what "Jewish state" actually means. Is Israel a state with Jewish religious characteristics or a secular state for the Jewish people? Israel was established as a democracy with a vision to be a national home and a safe haven for Jews, and the majority of its founders were secular Zionists who had not seen themselves bound to any rabbinical authorities or religious laws, although used religious rhetoric to describe their secular actions (Saposnik 2021). However, upon its creation, the Status Quo arrangements were established when David Ben-Gurion, Israel's first prime minister, promised the ultra-Orthodox leadership that the state would observe Shabbat as a day of rest for all Jews, official kitchens would be kosher, the Chief Rabbinate would be in charge of marriage and burial, and the Orthodox educational systems would remain independent. Ben-Gurion saw the secular and the Orthodox Jewish communities living side by side in peace, based on tolerance and mutual respect (Kedar 2013; Don-Yehiya 1989).

Demographic shifts in Israel, with a large influx of traditional Jews from the Muslim countries in the 1950s–1960s and high birthrates of Orthodox Jews, changed the secular socialist hegemony of the young state. From the 1970s forward, these trends also found a political voice with the rise of the Likud Party, which became the most dominant power in Israeli politics since 1977. Many Likud voters identify as *Masoratim* (traditionalists), and the Likud also built an almost unbreakable alliance with Orthodox parties (Leon 2016).

When it comes to the religious nature of the Jewish state, complex trends govern Israeli society. Scholar Guy Ben-Porat argues that, since the 1980s, there has been a trend toward *chilun* (secularization) in various aspects of Israeli society. This includes changes in religious observance, attitudes towards religious authority, and the role of religion in public life (Ben-Porat 2013). However, an opposing trend of *hadata* (religionization), a penetration of religious discourse into Israel's secular civil society, also exists. Yoav Peled and Horit H. Peled argued that this trend of religionization has been accelerating since 2000, and is manifested in a number of key social fields: the military, the educational system, the media of mass communications, the *teshuvah* movement, the movement for Jewish renewal, and religious feminism (Peled and Peled 2019). Shmuel Rosner and Camil Fuchs argue that both trends happen at the same time, and, while the forces in the Israeli establishment are promoting religionization, Israeli society is moving toward secularization (Rosner and Fuchs 2018).

The Paradox of Liberation, according to scholar Michael Walzer, is that secular liberation movements often result in unexpected religious revivals. He explores how revolutionary leaders in countries like India, Israel, and Algeria underestimated the profound cultural influence of religion, which reasserted itself after their secular successes, creating a paradox where secular revolutions unintentionally provoke religious counterrevolutions, challenging the movements' original goals (Walzer 2015). Scholar Yaacov Yadgar says that there is a secularist panic over this alleged "religionization", that is steaming from the demise of the Israeli left (Yadgar 2020, pp. 112–50).

The academic study of the religiosity of Israelis is growing, but studies based on survey data are limited. Rosner and Fuchs (2018) published the most rigorous study on the Jewish character of the State of Israel: *#Israeli Judaism: Portrait of Cultural Revolution.* Their research was based on a survey of 3005 respondents, conducted in two stages between 2017 and 2018.

Rosner and Fuchs argue that a unique blend of Judaism has developed in Israel. This blend is founded on the accelerated decline of halakhic Israeli identity, where more ultra-Orthodox are integrating into Israeli society; a decline in support for the religious–Zionist movement; a decline of traditionalist Israeli identity (i.e., many Israelis feel less comfortable identifying as traditionalists because they perceive this category to be antiquated and linked to ethnic identity); and, lastly, a decline in secular Israeli identity (Rosner and Fuchs 2018, pp. 22–31). These changes, argue Rosner and Fuchs, allow for a new phenomenon to emerge: "a mixing of Jewishness and Israeliness". Rosner and Fuchs dub them "Jewsraelis", practitioners of a new "IsraeliJudaism", which reflects a mixing of Jewishness and Israeliness. The authors show that they now comprise 55 percent of the Israeli Jewish population.

In their extended research, Rosner and Fuchs examined many aspects of Jewish identity and practice, and their research is essential. However, in their study, the authors have not addressed questions of religious beliefs. We seek to fill this gap by taking a somewhat different approach. Instead of asking about the levels of practice of religious rituals (like, for example, the celebration of holidays), we asked about the intensity of belief in certain core Jewish ideas. Thus, the survey allows us to unpack another layer in the complex views of Israeli Jews toward their Judaism.

## 2. Hypotheses

When it comes to religion, Israeli society is divided into three major blocs: Secular, *Masorati* (traditional), and Orthodox. Each of these blocs is not fully unified or coherent, and each has a spectrum of views represented within. Rosner and Fuchs show that, in their 2017–2018 surveys, 28% identified as totally secular, 21% somewhat traditional secular, 19% traditionalist, 5% liberal–religious, 10% national–religious, 7% dati-torani (also known as Hardal—nationalist Haredi), and 9% Haredi (ultra-Orthodox). If divided according to blocs: 28% are secular, 40% are traditional, and 31% are Orthodox.

In the last few decades, the traditional label was eroded when the Shas Party became the political representative of this community and has taken it into ultra-Orthodox religiosity. Although not all who support Shas are Orthodox, their "middle of the road" style of religiosity has shifted more into the Orthodox style (Leon 2016). Thus, the label "traditionalist" is complex. Rosner and Fuchs divided it into two sub-categories: 'Somewhat traditional secular' and 'traditionalist'.

It is common to divide Israeli Jews into a secular–religious binary. According to Rosner and Fuchs (2018), and also Pew Research Center's (2016) *Israel's Religiously Divided Society*, after dividing the traditionalists into sub-groups, Israeli Jewish society is split almost evenly between these two categories.

No similar statistical surveys have been conducted on the level of support for religious beliefs and eschatology among Israeli Jews, so it is hard to speculate based on past knowledge. However, since half of Israeli Jewish society is traditional and Orthodox, it

is reasonable to assume that this group would agree with traditional Jewish core beliefs, including the Election of Israel, the afterlife, and the messiah.

The 'secularization thesis' has been a prevailing theory in the study of religion and has argued that 'religion' and 'secular' are two opposite categories (Asad 2003; Bellah 1991; Berger 1996). However, 'post-secular' theories challenge the conventional understanding that religion and secularism are mutually exclusive dichotomies. Thus, a person can be religious and secular at the same time (Tylor 2007). Therefore, the label 'secular' does not necessarily mean agnostic or atheist. For example: The study of American religiosity showed most Americans (90%) believe in the existence of a higher power. According to a Pew Research Center study from 2017, only 10% of the American population say that they do not believe in God as described in the Bible or some level of a higher power; thus, only 10% of Americans are agnostics or atheists (O'Reilly 2018). Scholar Hagar Lahav argues that about half of Israeli seculars are 'secular believers'—a hybrid identity that includes a lot of ambivalence. Judaism is mostly a religion of practice. Lahav says that secular believers hold a deep sense of Judaism and faith alongside a highly selective attitude toward Orthodox teaching and practice (Lahav 2017; see also Jobani 2008, 2016). Returning to Israeli Jews' beliefs, we can speculate that most Israeli Jews, including those identifying as 'secular,' might believe in some type of higher power, as we have seen in America.

The Election of Israel comprises two premises: First is that Jews are the "Chosen People" of God, and, as such, they are entitled to "The Promised Land". The secular Zionist idea of negation of exile and normalizing the Jewish people rejects, on an ideological level, the concept that Jews are different from any other nation. The Zionist ideology argued that the Jewish state would be a normal state, like all others. Thus, it is expected that secular Israeli Jews would be hesitant toward the biblical notions of Jewish election and chosenness. However, the Zionist narrative argues that Jews are entitled to a state based on the ideas of "ingenuity" and that Jews are the heirs of their ancient homeland. The borders of the ancient Land are based on biblical promises. We can assume that the traditional–Orthodox block would agree with the ideas of the Abrahamic Covenant, but the secular block would have more mixed feelings. From a secular Zionist perspective, we can assume that many would agree with the notion of the Promised Land (although not necessarily the ideas of Greater Israel) but would support less the "chosen people" component of the Covenant.

When it comes to collective salvation and the belief in the coming of the messiah, we can assume that those who are traditional–Orthodox would strongly believe in it, while those who identify as secular might be less inclined or even oppose it altogether. The secular Zionist ethos stood as a rejection of the passive messianic expectations of the generations of Jewish exile. Zionism was a rebellion against the Jewish waiting for the messiah. Also, to be "messianic" in Israel has a unique meaning, especially among left-leaning circles. The label applies to those who support the religious–Zionist ideology of "Greater Israel" that views Israeli reality as part of a messianic process. The settlement enterprise in the West Bank that was promoted by the religious–Zionist activists is often referred to as motivated by messianic beliefs or pushing the End of Times, and not all parts of Israeli society support the ideological settlers. It should be mentioned that many of the settlers were not motivated by religious ideologies but were drawn to the settlement for inexpensive housing opportunities and a better quality of life (Gorenberg 2006).

Zionist pioneers in the early days of Jewish settlement in the Land rebelled against the passivity of the traditional circles regarding immigration to the Land of Israel. Although using messianic language to describe their secular activities, they overwhelmingly rejected the messianic tradition of past generations that only the messiah, at the End of Days, could deliver the people of Israel from their exile to their messianic redemption (Troen 2024).

This passive tradition was one of the reasons why ultra-Orthodox Judaism was reluctant to join the Zionist enterprise (Ravitzky 1993). Not all Orthodox Jews support the link between the state and the coming of the messiah; however, since there is a rooted Zionist tradition that identifies the messianic expectations as something negative that is holding back from national realization, we can expect that those who identify as secular might be

negatively predisposed to the principled idea of the coming of the messiah from a national perspective that negates religious passivity. We can also expect that those who reject the idea of the messiah because of historically secular Zionist ideology might be from the older generations. The founding Zionist ideology was associated with the Socialist Labor Party (Mapai), but, since 1977, the Likud party has dominated the Israeli political sphere. The Likud is a right-wing, traditionalist, capitalist party. Therefore, younger Israeli Jews might be less influenced by left-wing secular historical Zionist ideologies.

Jewish tradition anticipates that, with the coming of the messiah, ancient institutions would be restored: The Davidic kingdom, the Temple on the Temple Mount, and the Sanhedrin as the supreme court. In 1967, Israel took the Temple Mount from Jordanian control during the Six-Day War, and, theoretically, the State of Israel can build a temple for God on the Temple Mount. Such an idea has never come to pass. After the Mount was taken by the IDF from the Jordanian Legion, Israel initiated the Status Quo Arrangements that declared the Temple Mount as an exclusive Muslim worship site. However, tourists are allowed to visit it, while the Western Wall was to become an exclusive Jewish worship site (Bar and Cohen-Hattab 2021; Berkovitz 2001). Most Jews agreed with these arrangements when they were instituted for multiple reasons: The Chief Rabbinate made a clear statement during the war that Jews are not allowed to visit the Temple Mount because there are purification rituals that currently cannot take place with the absence of a red heifer. Most Orthodox rabbis agreed with this ruling (Inbari 2009). Also, the Mount is considered the third holiest site for Islam, and many Israelis prefer not to provoke the Muslim world (Berkovitz 2001). However, since its occupation, there was a minority voice that called to allow Jews to pray on the site, to take control of the site from the Jordanian Waqf, and to make it a national symbol (Hershkowitz 2022). After the Israeli Disengagement from Gaza in 2005, growing voices within the national religious movement are calling to allow praying on the Mount. It is a growing trend, and, in 2023, it was reported by activists that 50,000 Jews visited the holy site, most of them are young religious Zionists, demanding not only to be allowed visits as the Status Quo arrangements say but also to be allowed to openly pray on the site. Currently, they are only permitted silent prayer (Gross 2022).

When it comes to the attitudes toward the Temple Mount, we can assume that, for those identifying as secular, visiting the site would not be considered controversial because it is part of the Status Quo Arrangements with the Waqf, but praying on the site would be favored by mostly religious–Zionists, while the ultra-Orthodox would oppose it, due to religious reasons of ritual purification. Although the Temple Mount is the holiest site for Judaism, because of the Status Quo arrangements, this is the only place in Israel where Jews are not allowed to pray, and there is a segment of Israeli society that is outraged by that (Persico 2017; Fischer 2017). Although not necessarily Orthodox, they might support praying on the site from a nationalist perspective.

In conclusion, whereas we can expect that those who identify as Orthodox and traditional would support ideas of personal and collective redemption, those who identify as secular would be less inclined, but the case of the Temple Mount breaks this binary. We can assume that some Orthodox circles, especially religious Zionists, would support ritual on the Temple Mount from a nationalistic or messianic perspective, while other Orthodox circles would oppose it from a purification perspective; some seculars would support it because of nationalistic reasons, while other seculars would oppose it in order not to create provocations with the Palestinians.

This article will begin by providing a demographic overview of our sample and then pivot to presenting the religious scale of Israeli Jews. From there, we review the survey's results regarding belief in God, support for statements relating to the Election of Israel, personal salvation, collective redemption, and the Temple Mount. The article ends with a discussion that places our findings within the context of broader research on these important subjects. The article ends with a speculation of how the 7 October 2023, and the outbreak of the Israel–Hamas war, shifted Israeli public opinion on religious matters.

### 3. The Survey

Between 26 March and 18 April 2023, we conducted a survey among a representative sample of 1204 Israeli–Jewish respondents. The sample provides a 95% confidence level that the sampling error does not exceed ±2.82%. Respondents were interviewed online and over the phone in Hebrew. The sample includes 1004 online respondents and 200 phone respondents (for those who do not use the Internet, such as people over 65 years old and some ultra-Orthodox respondents). GeoCartography (Tel-Aviv, Israel) administered the survey on our behalf.[1] The survey's timing was after the coronavirus lockdowns, after five election cycles in four years, and during a growing tension in Israel over the judicial reform legislation. During that time, Netanyahu's coalition of Likud and Orthodox parties attempted to legislate laws that would weaken judicial power and was confronted with weekly mass demonstrations. The tension of that time, with much internal political polarization, might have sharpened the views of the respondents. In comparison to the 2017–2018 surveys conducted by Rosner and Fuchs, we saw, in our data, that Israelis are more likely to identify as 'completely secular'; so, the heightened level of polarization in Israeli society in the last few years permeates our data (see later in the article). The survey was administered six months before the tragic events of 7 October 2023, and the war that broke out between Israel and Hamas.

In our survey, 48% were men, while 52% were women. Age distributions were 18–29 years old at 29%, 30–49 years old at 37%, 50–64 years old at 20%, and 65 and older at 14%. The survey's results largely mirrored the 2022 election results, with only slight deviations. In the survey, 62% of Israeli Jews identified as right-wing, about 19% as centrist, while left and center–left at around 14%. The remaining 5% did not provide a response.

In order to examine the religious labels of Israeli Jews, we followed a similar scale to that of Rosner and Fuchs from their 2017–8 research. We added two additional categories: Reform–Conservative (4.5%) and Other (0.3%). The addition of these two labels has had little impact on the results (see Figure 1 below).

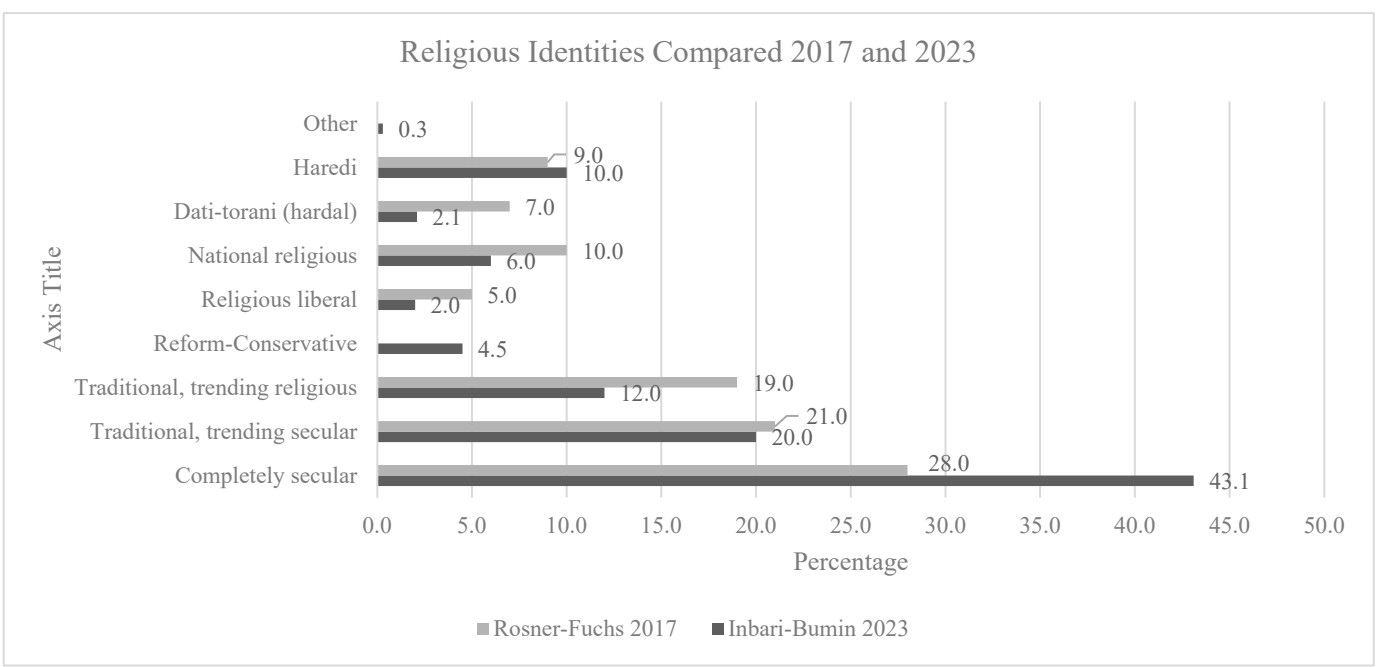

**Figure 1.** Religious identities comparing 2017 and 2023.

The 2017–2018 survey was taken during relatively stable political and national times, while the 2023 survey was taken after a period of political instability, with multiple indecisive general elections, and during a period of tensions and growing anti-government demonstrations. While comparing the two surveys, a few differences are striking regard-

ing the polarization of the Jewish society: between 2018 and 2023, the secular label has increased, while traditional and religious Zionist (religious liberal, national religious, and dati-torani) have decreased in numbers.

We added another question to examine the religiosity of Israelis: "How often do you pray?" The results that appear in Figure 2 show that 53.6% never or seldom pray, while 46.4% pray at least once a month.

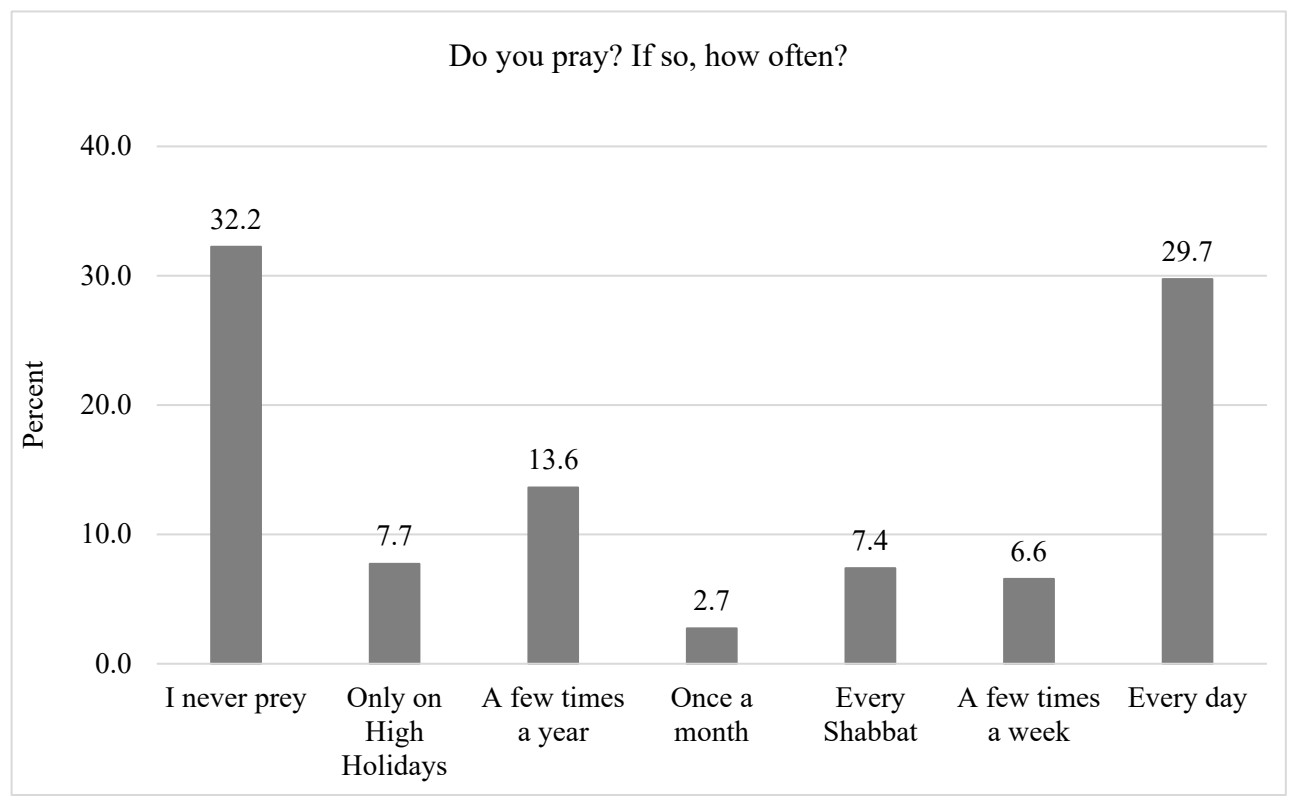

**Figure 2.** Prayer.

Putting self-identification and prayer level together shows that the secular–religious binary in our survey is similar to Rosner and Fuchs's (2018) results and that Israeli Jewish society is split almost in half along these lines; there is almost a perfect correlation between how Israelis identify and their frequency of praying.

As Figure 3 illustrates, cross-tabulation of prayer and party vote illuminates some interesting trends. As expected, about 70% of those who voted for Orthodox parties (Shas, Yahadut Hatorah, Hatzyonut Hadatit, and Otzma Yehudit) pray every day (Orthodox women do not have to pray daily, but many do). While 70% of those who voted for left-wing parties (Meretz and Haavodah) never pray. A total of 80.1% of those who voted for centrist parties (Hamachane Hamamlachti and Yesh Atid) also seldom or never pray. There is clearly a robust correlation between voting patterns and religiosity among our respondents. The only exception is Likud voters, who are spread among all the categories, with about 51% saying they never or seldom pray and 49% reporting they pray at least once a month.

Zooming into the prayer patterns of supporters of Orthodox parties (see Table 1 below) shows that voters of Hatzyonut Hadatit and Otzma Yehudit pray the least of all Orthodox parties, meaning that this party has attracted about 30% of its voters from outside Orthodox or traditional circles. Also, Shas attracts outside the Orthodox/traditional circles, but less than 10% of its voters say that they never or seldom pray. These are the only parties that can significantly recruit outside their own immediate circles.

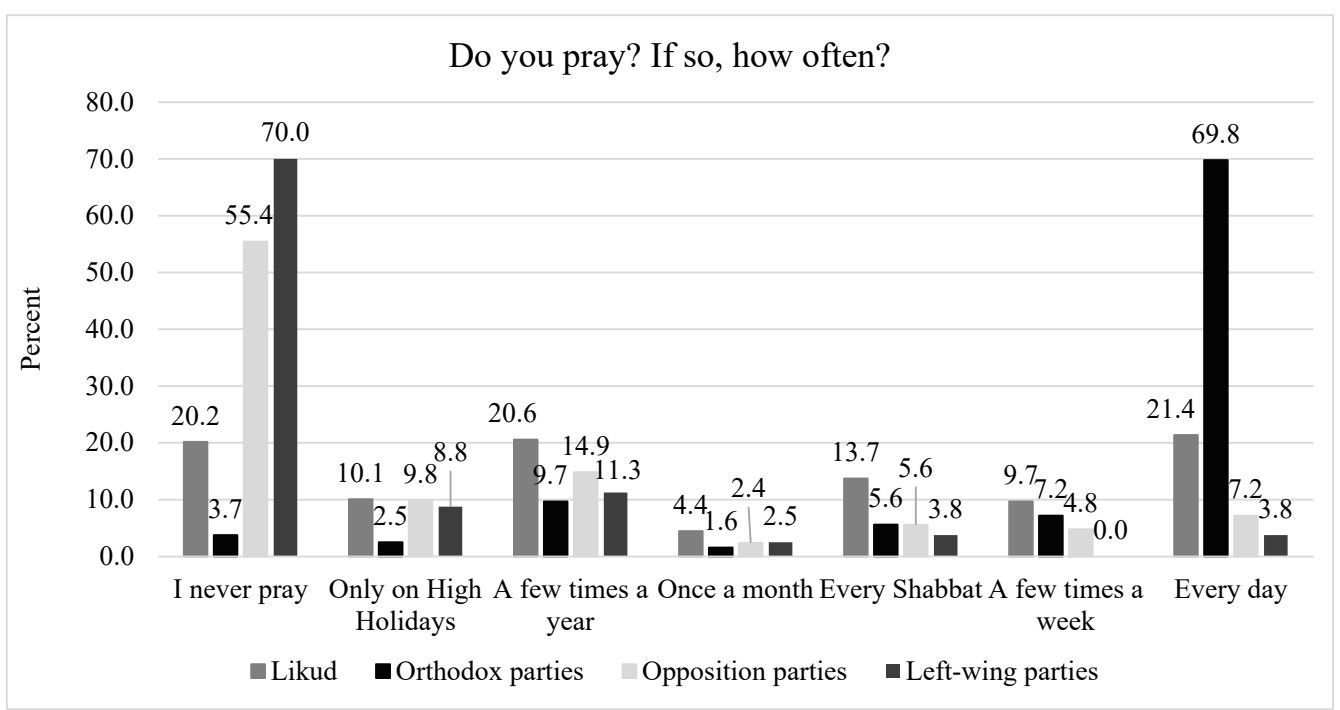

**Figure 3.** Prayer and voting patterns.

**Table 1.** Prayer and voting pattern for Orthodox parties.

|  | Likud | Shas | Yahadut | Hatziyonut |
|---|---|---|---|---|
| I never pray | 20.2% | 1.4% | 0% | 7.6% |
| Only on High Holidays | 10.1% | 2.7% | 1% | 3.5% |
| A few times a year | 20.6% | 5.4% | 1% | 18% |
| Once a month | 4.5% | 0% | 0% | 3.5% |
| Every Shabbat | 13.7% | 0% | 2% | 11.1% |
| A few times a week | 9.7% | 8.1% | 5.8% | 7.6% |
| Every Day | 21.4% | 82.4% | 90.3% | 48.6% |

*3.1. God*

Does identifying as "secular" necessarily mean denying the existence of God or denying fundamental Jewish beliefs about the Abrahamic covenant? Based on a modified Pew question about the belief in God, we gave the participants three options, as described below in Table 2.

**Table 2.** Belief in God.

| Statements | Percentage |
|---|---|
| I believe in God as described in the Jewish tradition | 63.6% |
| I believe there is some other higher power or spiritual force in the universe but I do not believe in God as described in the Jewish tradition | 22% |
| I do not believe there is ANY higher power or spiritual force in the universe | 14.4% |

As we can see, almost 86% of Israeli Jews believe in some type of higher power. We can identify additional trends when we cross the beliefs in God and reported income. Apparently, there is a correlation between income and beliefs: the less prosperous Israelis

are more religious, and vice-versa.[2] Those who believe in the higher power option tend to belong to above-average income categories.

We can conclude that most Israeli Jews believe in the biblical God or a higher power, and these beliefs are influenced by demographic aspects of income (as Figure 4 shows) and, as we discuss later, in terms of age, younger cohorts express higher levels of religiosity than older Israelis.

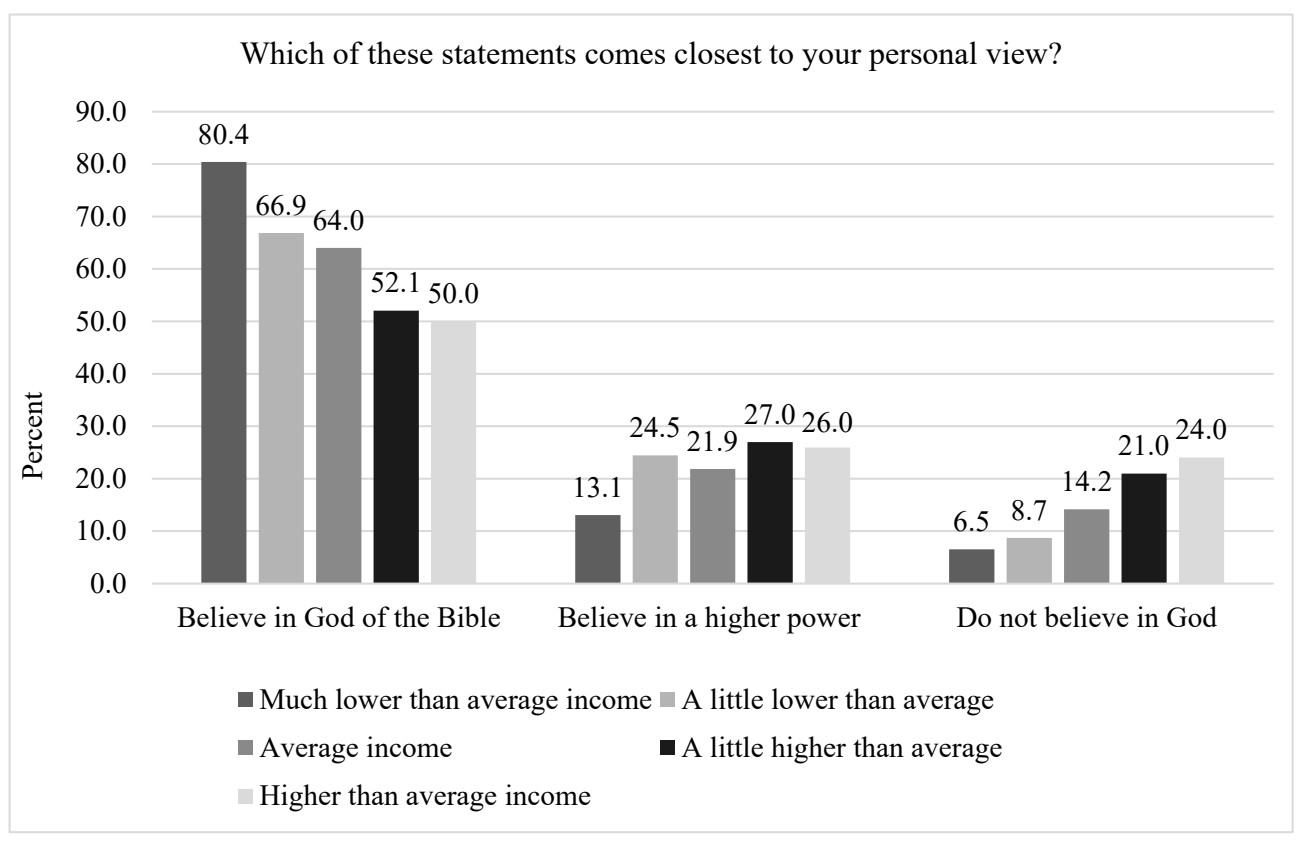

**Figure 4.** Belief in God/higher power by income.

A two-way tabulation of belief in God and prayer shows that, as expected, most of those who do not believe in any type of higher power also never pray, but this group is only 39% of the sample, while 24% who never pray believe in the biblical God and 37% believe in a higher power. Although they never pray, about 60% of them still hold some beliefs about God. Among those who pray, even seldomly, the levels of not believing in God drop to less than 10%. Thus, we can see that even those who identify as completely secular hold diverse views about God or a higher power (this will be discussed at greater length below).

### *3.2. Covenant*

The Bible says there is a covenant between Abraham and God, making his descendants a chosen people who are promised a certain land called The Land of Israel (for example, see Genesis 12: 1–3).

In the survey, we asked the respondents if they agree with these biblical statements, which also carry a political meaning in The State of Israel.

The response to these statements, which appears in Table 3, shows that there is strong support among Israeli Jews for the ideas of the Abrahamic Covenant. Another question that examines the religiosity of Israeli Jews asked the respondents to express their opinion on the following statements.

**Table 3.** Abrahamic covenant.

|  | Yes | No | I Don't Know |
|---|---|---|---|
| Do you believe Jews are the 'Chosen people' as the bible describes? | 64.2% | 22.4% | 13.4% |
| Do you believe that the Land of Israel is 'The Promised Land' as the bible describes? | 73.9% | 15.6% | 10.5% |

From the Israelis' responses (see Table 4), we can see the consistency that about half of Israeli Jews are traditional and Orthodox, viewing Judaism as the only true religion.

**Table 4.** Beliefs of Israelis.

|  | Percentage |
|---|---|
| I believe in spirituality but not in any formal religion | 14.4% |
| I believe in the Jewish religion and it is the only true religion | 50.4% |
| I believe in the Jewish religion but it is not the only true religion | 16.6% |
| I believe in a different religion which is not the Jewish religion | 0.3% |
| I don't believe in God | 11.9% |
| Don't know/none of them | 6.4% |

It becomes more complicated as we dig deeper into what "secularity" actually means. There are two ways to examine how those who identify as secular responded to these faith-based questions. In Table 5, readers can see how those who believe in the Jewish religion but do not think it is the only true religion, those who are spiritual but not religious, and those who do not believe in God responded to the previous questions. They will be followed by those who identified as 'secular' or 'traditional, trending secular'.

**Table 5.** Seculars and beliefs.

|  | Belief in Biblical God | Belief in Some Higher Power | No Belief |
|---|---|---|---|
| Not the only religion | 59.5% | 37% | 9.5% |
| Spiritual | 23% | 61% | 16% |
| No God | 0 | 15% | 84% |
| Completely Secular | 28% | 39% | 33% |
| Traditional, trending secular | 71% | 23% | 6% |
| "Chosen People" | Yes | No | I don't know |
| Not the only religion | 55.5% | 20% | 24.5% |
| Spiritual | 29% | 45.5% | 25.5% |
| No God | 6% | 84% | 10% |
| Completely Secular | 31% | 47% | 22% |
| Traditional, trending secular | 68.5% | 13.5% | 18% |
| "Promised Land" | Yes | No | I don't know |
| Not the only religion | 75.5% | 9.5% | 15% |
| Spiritual | 50% | 30% | 20% |
| No God | 21.5% | 66.5% | 12% |
| Completely Secular | 45.5% | 35% | 19.5% |
| Traditional, trending secular | 83.5% | 6% | 10.5% |

We can see that 60–75% of those who say they believe in the Jewish religion but do not think it is the only true religion (16.6% of the total sample) mostly agree with the traditional statements. Those who identify as spiritual but not religious (14.4%) confess that they hold some level of faith but tend to be more critical of these traditional statements, while those who say they do not believe in God (11.9%) are the most critical. Comparing them with those who identify as 'secular' or 'traditional, trending secular' shows some overlap in the responses between those who identified as 'spiritual but not religious' and those who identified as 'secular.' There is a smaller overlap between those who identified as 'traditional, trending secular' and those who say they 'believe in the Jewish religion, but it is not the only true religion.'

To conclude, the Israeli Jewish society is divided into binary secular–religious labels but digging deeper into the meaning of these labels shows that, in the Israeli context, "religious" means some type of Orthodox or traditional religiosity, while "secular" respondents profess a spectrum of beliefs from traditional to atheism. Most of them are indeed 'secular-believers' of some sort.

### 4. Eschatology: Personal Salvation

The belief in the afterlife is central to Jewish teaching and derives from the belief in reward and punishment. The eighteen Benedictions, the most fundamental Jewish prayer, include a prayer for the restoration of the dead, and Maimonides' Thirteen Principles of Jewish Faith include: "I believe with perfect faith in the coming of the messiah, and though he may tarry, still I await him every day; [and] I believe with perfect faith that there will be a revival of the dead at the time when it shall please the Creator, Blessed be His name, and His mention shall be exalted forever and ever".

In the survey, we asked multiple questions about life after death, and these results are presented in Table 6 below.

**Table 6.** Personal redemption.

|  | Yes | No | I Don't Know |
| --- | --- | --- | --- |
| Do you believe that there is life after death? | 49.3% | 28.2% | 22.6% |
| Do you believe that there is a world to come (like heaven and hell)? | 54.5% | 26.3% | 19.2% |
| Do you believe in reincarnation? | 48.6% | 28.6% | 22.8% |

In addition, we asked: "Do you believe that one day there will be a "messianic age" of peace and prosperity?". Those who agreed to this question were then asked a follow-up question: "Do you believe in the resurrection of the dead in the End Times?". This data appears below, in Table 7.

**Table 7.** Messianic age.

|  | Yes | No | I Don't Know |
| --- | --- | --- | --- |
| Do you believe that one day there will be a "messianic age" of peace and prosperity? | 48.8% | 33.6% | 17.5% |
| Do you believe in the resurrection of the dead in the End Times? (asked only among those who agreed with the preceding statement) | 77.6% | 6.5% | 15.9% |

Around 50% of the sample agrees with all of these traditional beliefs. The second half is split between disagreement and "I Don't Know".

In the survey, when we cross-tabbed many questions, we saw distinctive views among different age groups. As Figure 5 illustrates, the youngest cohort (18–29-year-old respondents) exhibited the highest levels of agreement with the belief in the afterlife, while the

oldest respondents exhibited the least support. An example is in the responses about beliefs regarding the "world to come (like heaven and hell)".

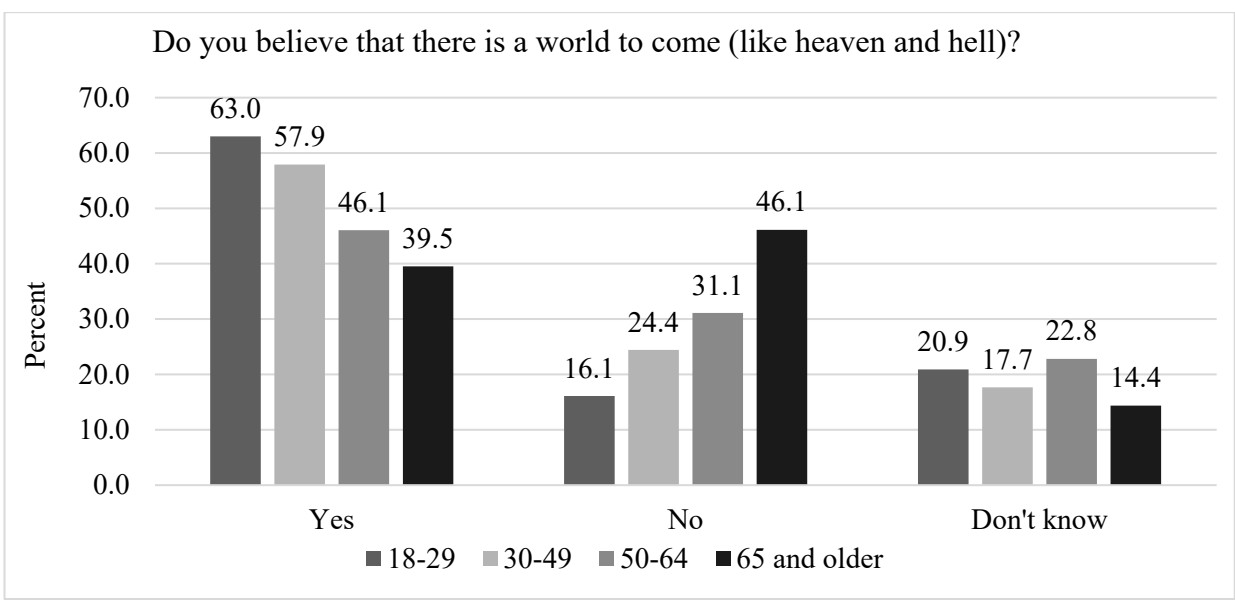

**Figure 5.** Beliefs in heaven and hell ("World to Come") across age groups.

Another way of looking at the results is via party vote (see Figure 6 below). As the graph below shows, the supporters of the idea of the world to come mostly come from voters of Orthodox parties and the Likud, while most who oppose them are voters of centrist opposition parties and left-wing parties.

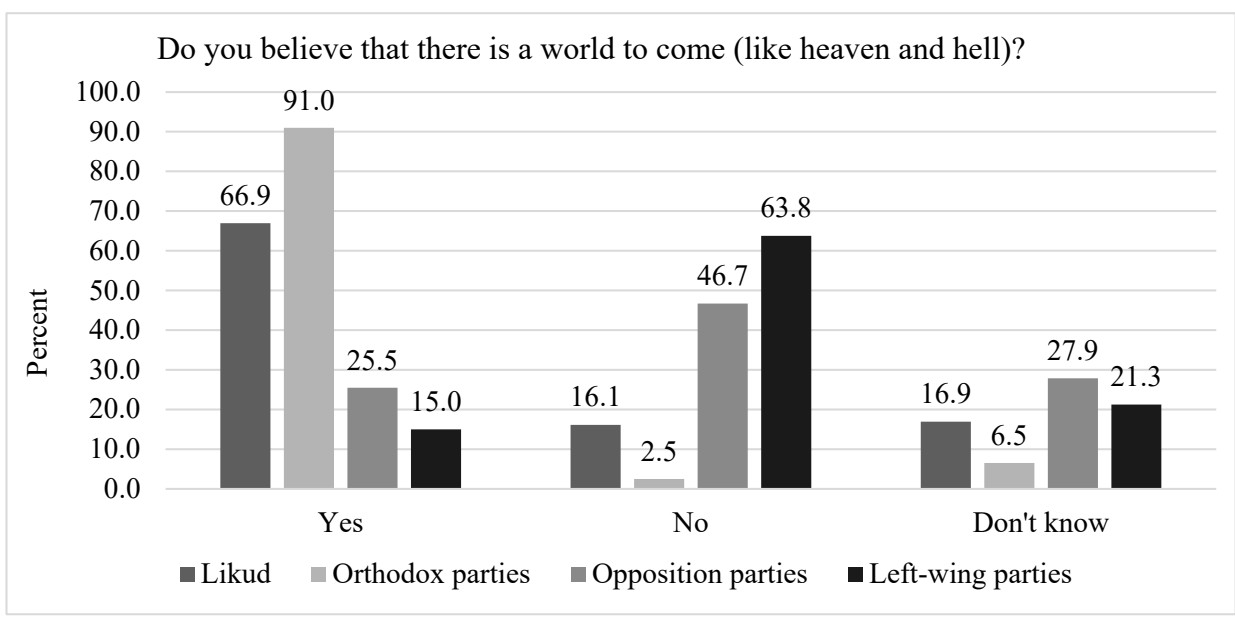

**Figure 6.** Beliefs in heaven and hell ("World to Come") by party vote.

From the results of the survey, we conclude that, as expected, close to half of the respondents agree with beliefs of personal salvation and life after death, as rooted in Jewish tradition. These general views have strong support among the younger generations and those with lower incomes. In political terms, voters of Orthodox parties and the Likud mostly support these traditional beliefs. A total of 25–30% of the sample exhibit opposition

to these ideas (older and wealthier are more prevalent in this subgroup). About a quarter or less of the sample chose the "I don't know" option, demonstrating unwillingness to decide.

Among the secular respondents, we can see a large "I don't know" group, which can be understood as uncertainty toward the ideas of personal salvation.

In Table 8, we can see that those identifying as secular tend to oppose the traditional views of the afterlife and the world to come, while those who identify as 'traditionalist trending secular' tend to agree with them. The picture becomes more complex when we examine the three categories of belief and the questions regarding the unseen: those who 'believe in the Jewish religion but it is not the only true religion' and 'spiritual but not religious' groups have their views divided between the three options pretty equally.

**Table 8.** Afterlife.

| Do You Believe in the Afterlife? | Yes | No | I Don't Know |
|---|---|---|---|
| Judaism is not the only religion | 30% | 37% | 33% |
| Spiritual | 34% | 34% | 32% |
| No God | 5% | 80.5% | 14.5% |
| Secular | 22% | 49.5% | 28.5% |
| Traditional, trending secular | 46% | 16% | 20% |
| **Do You Believe in the World to Come?** | **Yes** | **No** | **I Don't Know** |
| Judaism is not the only religion | 35% | 29% | 36% |
| Spiritual | 30.5% | 41.5% | 28% |
| No God | 2% | 87% | 11% |
| Secular | 21% | 52% | 27% |
| Traditional, trending secular | 53% | 21% | 26% |

## 5. Collective Redemption

The belief in the coming of the messiah is essential in Judaism. With the rise of the Zionist movement, Orthodox circles have debated whether the new movement was a type of messianic realization. Some Orthodox rejected this connection (Keren-Kratz 2024, pp. 28–45), while others accepted it (Schwartz 2002). Similarly, secular Zionists contemplated this question and pondered whether their actions were a type of secular messianism (Saposnik 2021; Ohana 2009).

We asked a simple question in our survey, "Do you believe in the coming of the messiah?", and gave the respondents only "yes" and "no" options. A total of 55% of the surveyed Israelis believe in the coming of the messiah. The respondents' age, income, and political vote impacted how they responded to this question. Here is an example:

As Figure 7 illustrates, the youngest Israeli Jews are the most supportive age cohort of this notion, while the oldest respondents are the least supportive.

For those who believe in the coming of the messiah, we wanted to examine how strong of a connection they see between the State of Israel and the messiah. This connection taps into the religious Zionist narrative that the State of Israel represents the "beginning of redemption" and that there is holiness in the state itself (Mirsky 2014). We therefore followed up with the respondents who believe in the coming of the messiah by asking them if there is a connection between the State of Israel and the coming of the messiah.

The results in Table 9 show that, indeed, most of those who hold traditional beliefs in the messiah also draw the connection between contemporary Israel and future redemption.

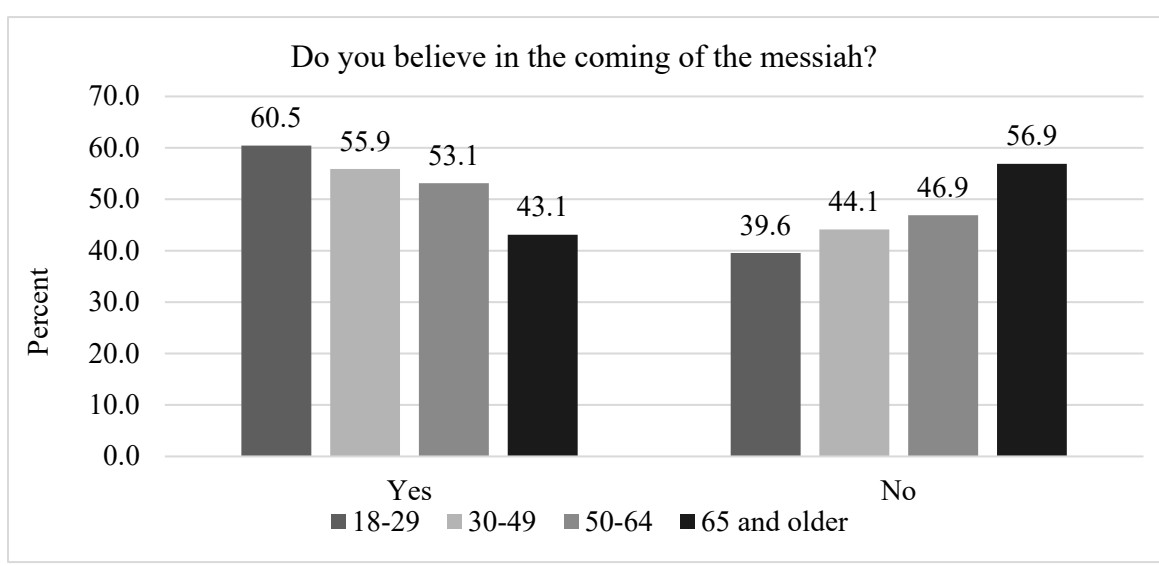

**Figure 7.** Age and the coming of the messiah.

**Table 9.** The messiah and the State of Israel (only for those who believe in the coming of the messiah).

|  | Yes | No | I Don't Know |
|---|---|---|---|
| Do you believe the messiah will perform miracles? | 68.8% | 6.1% | 25.1% |
| Do you believe there is a connection between the State of Israel and the coming of the messiah? | 63.7% | 20.1% | 16.2% |

Although the connection between the state and the messianic times is associated mostly with the religious Zionist sector and with Hatzyonut Hadatit and Otzma Yehudit voters, the data presented in Table 10 show that it is more widespread than one Orthodox sector. The breakdown of party votes and the belief that there is a connection between the State of Israel and the coming of the messiah find strong support among Likud, Hatzyonut Hadatit, and Shas voters.

**Table 10.** Party vote and connection between the state and the messiah.

|  | Yes | No | I Don't Know |
|---|---|---|---|
| Likud | 52.7% | 15.7% | 31.6% |
| Yahadut Hatorah | 14.4% | 62.7% | 22.9% |
| Hatzyonut/Otzma | 48.8% | 25.5% | 25.7% |
| Shas | 47.2% | 19% | 33.8% |

These results are interesting and somehow surprising. The religious–Zionist narrative of seeing a connection between the state and the messiah found support among only half of the Hatzyonut Hadatit voters but also among half of the ultra-Orthodox Shas and half of the Likud supporters. It means that this idea is more popular than just the niche of the party that represents the religious–Zionist circles. The ultra-Orthodox, non-Zionist narrative that denies this connection is prevalent only among Yahadut Hatorah voters, but, also, among them, there is some uncertainty toward this idea.

## 6. The Temple Mount

The belief in redemption also includes the idea of the restoration of the Temple in the End of Times. In the Amidah prayer (the Eighteen Benedictions), there are multiple references to the desire for redemption and the reconstruction of the Temple. Here is just

one example (translation by https://www.chabad.org (accessed on 4 September 2024)): "May it be your will, L-rd our G-d and G-d of our fathers, that the Bet Hamikdash (The Temple) be speedily rebuilt in our days".

As the quote above indicates, the messianic expectations regarding Jewish restoration are significant in Jewish prayer and memory. However, the restoration of the Temple has not been "on the table" among the mainstream Jewish society in Israel since 1967, when Israel took the site. Since its occupation, the Temple Mount has been open for tourism, but it is considered an exclusive worship site for Muslims. Since the late 1990s, a growing number of mostly religious Zionists has been demanding that Jewish prayer be allowed on the Mount.

Given the increasing importance of this issue, our survey asks what Israeli Jews think about visiting the site and restoring Jewish worship on it, now or at the End of Days.

Our first question, presented in Table 11, was about Jewish visits to the Mount. As a reminder, according to the Status Quo arrangements, tourists are allowed to visit the site.

**Table 11.** Visits to the Temple Mount.

|  | Yes | No | I Don't Know |
| --- | --- | --- | --- |
| Do you support Jews visiting the Temple Mount? | 38.2% | 41.9% | 19.9% |

As the results show, there is almost an even split in opinions, with 20% indecisive. As we expected in our hypothesis, the binary secular–religious responses have not applied to this question, as people may develop an opinion on this question based on multiple parameters: national pride, ritual purification restrictions, and Jewish–Muslim relations.

It is interesting to see how the respondents' self-reported voting patterns influenced their choices on this question. Table 12 presents the complete list of responses:

**Table 12.** Support for Jews visiting the Temple Mount divided by party vote.

|  | Shas | Yahadut Hatorah | Hatziyonut Hadatit | Yisrael Beitenu | Likud | Yesh Atid | Hamachane Hamamlachti | Haavoda | Metetz | Hadash |
| --- | --- | --- | --- | --- | --- | --- | --- | --- | --- | --- |
| yes | 23.0% | 2.9% | 60.4% | 51.6% | 60.5% | 26.5% | 40.9% | 16.3% | 3.2% | 0.0% |
| No | 64.9% | 91.3% | 20.1% | 29.0% | 15.4% | 52.1% | 44.9% | 57.1% | 77.4% | 0.0% |
| I don't know | 12.2% | 5.8% | 19.4% | 19.4% | 22.2% | 21.5% | 14.2% | 26.5% | 19.4% | 100% |

From this table, we can see that only Meretz and Yahadut Hatorah voters are strongly opposed to visits to the Mount. A majority of Haavoda, Shas, and Yesh Atid voters oppose it as well, but opinion among these respondents is more evenly split. Thus, we can observe interesting coalitions: ultra-Orthodox Yahadut Hatorah and secular Meretz voters strongly oppose visiting the Mount, while the anti-clerical Yisrael Beitenu and the Orthodox Hatzionut Hadatit strongly support it. Centrist voters of Hamachane Hamamlachty are divided evenly over this question.

Those who agreed with Jews visiting the Temple Mount (460 respondents out of 1204) were asked a follow-up question: "Do you support Jewish prayer on the Temple Mount?". In the recent decade, the question of Jewish prayers on the site has been a "hot topic" since many religious–Zionist rabbinical authorities have softened their opposition to this ritual (Hershkowitz 2022). As Figure 8 below shows, almost all who agree with allowing Jews to visit the Mount also agree with Jewish prayer on the Mount—almost 87% agree. Thus, we can assume that, for many Israelis, visiting the Mount is understood as with the intention of praying on it. While visiting the site is part of the Status Quo agreements, Jewish prayer is not.

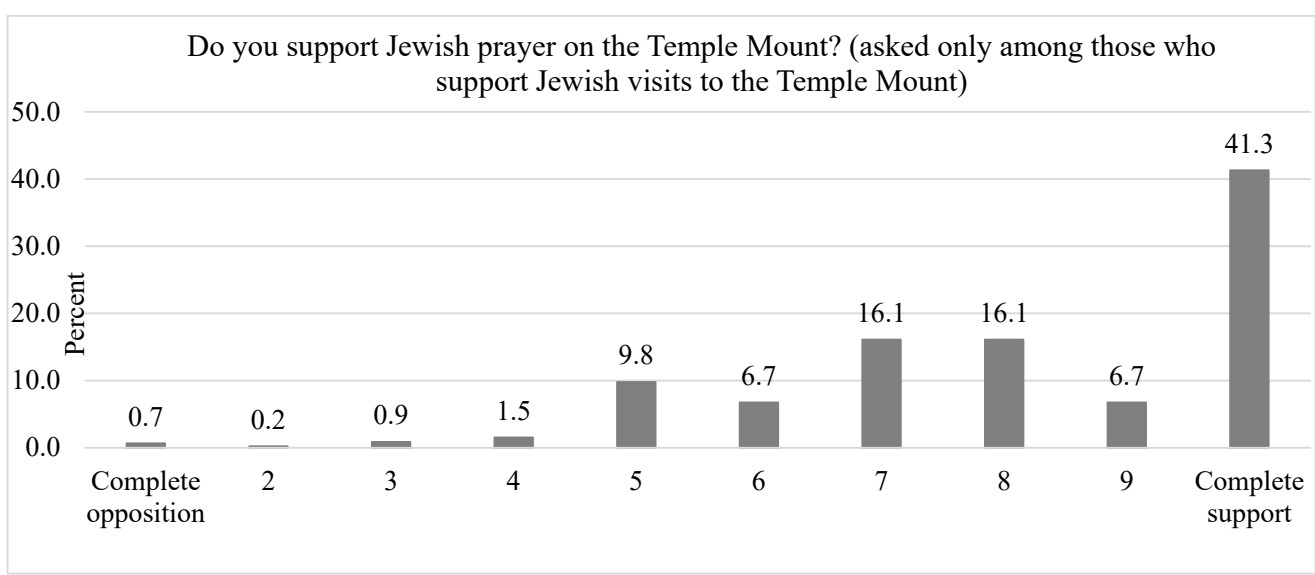

**Figure 8.** Praying on the Temple Mount.

We asked a follow-up question among those who would allow Jewish prayer on the Mount (400 respondents), asking them if they believe the State of Israel should build a synagogue for them to pray on the Temple Mount in the coming decades. Figure 9 presents the results of that question.

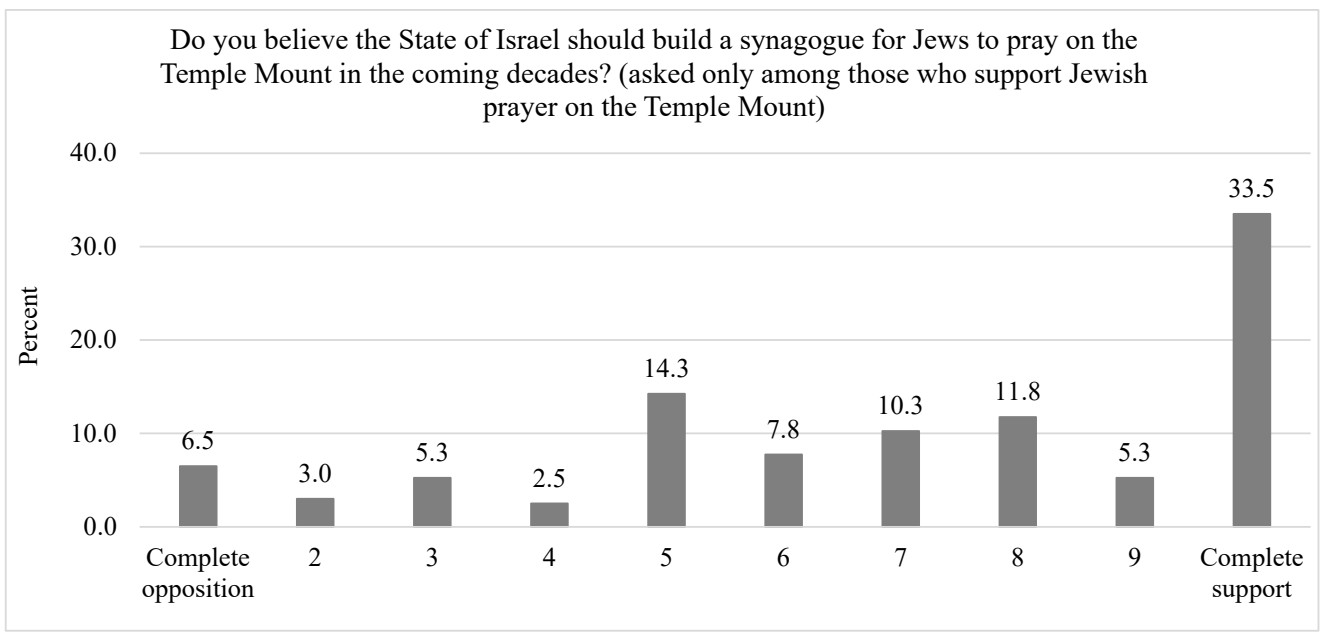

**Figure 9.** A synagogue on the Temple Mount.

The idea of building a synagogue on the Temple Mount is not new. Chief Rabbi Mordechai Eliyahu expressed it in 1991 (Inbari 2009, p. 24). Activists of Jewish presence on the Mount raise it from time to time.

A total of 17.3% of respondents strongly oppose it (1–4 responses on a 10-point scale), while 60.9% strongly support the idea of building a synagogue (7–10 responses). A total of 21.8% of respondents take a somewhat indecisive position (5–6 responses). Those who supported the idea of building a synagogue on the Temple Mount in the coming decades thus stand at around 20% of the entire sample.

Another set of questions we asked concerns the idea of a Third Temple. In Jewish tradition, it is assumed that a Third Temple would stand on the Mount as a manifestation of the End of Days. As quoted earlier, this belief is expressed daily in the Jewish prayer. The responses concerning the Third Temple are graphed in Figure 10.

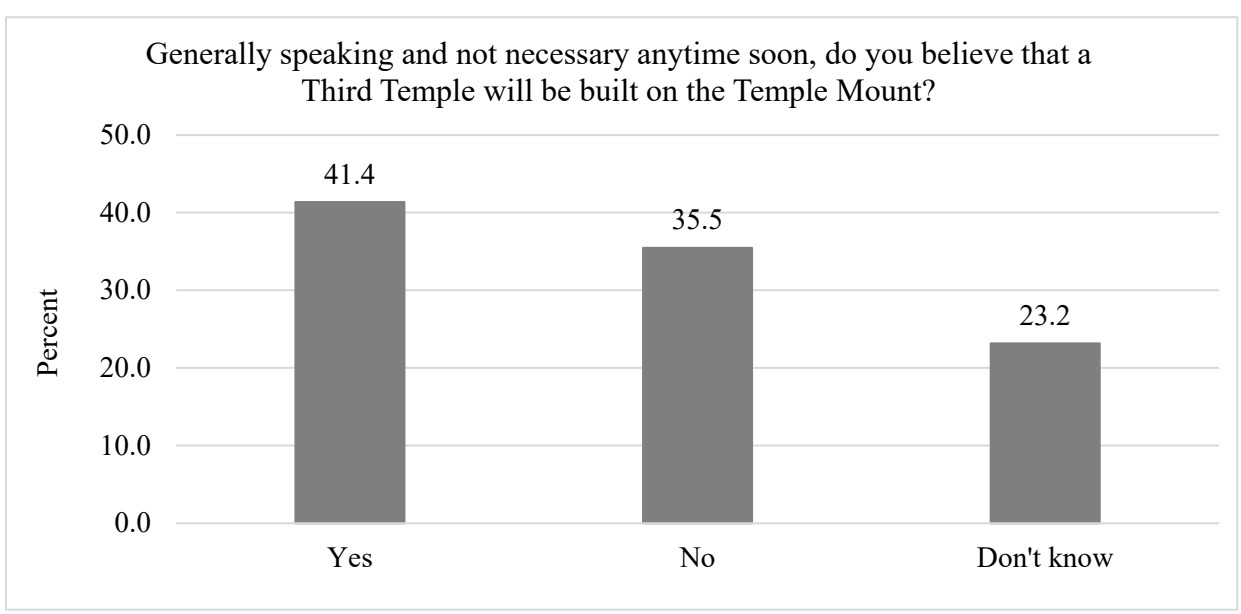

**Figure 10.** The Third Temple.

The breakdown of the results in Figure 11 below shows that the most significant demographic difference is age cohorts, with the largest support from the younger respondents and the strongest opposition from older generations.

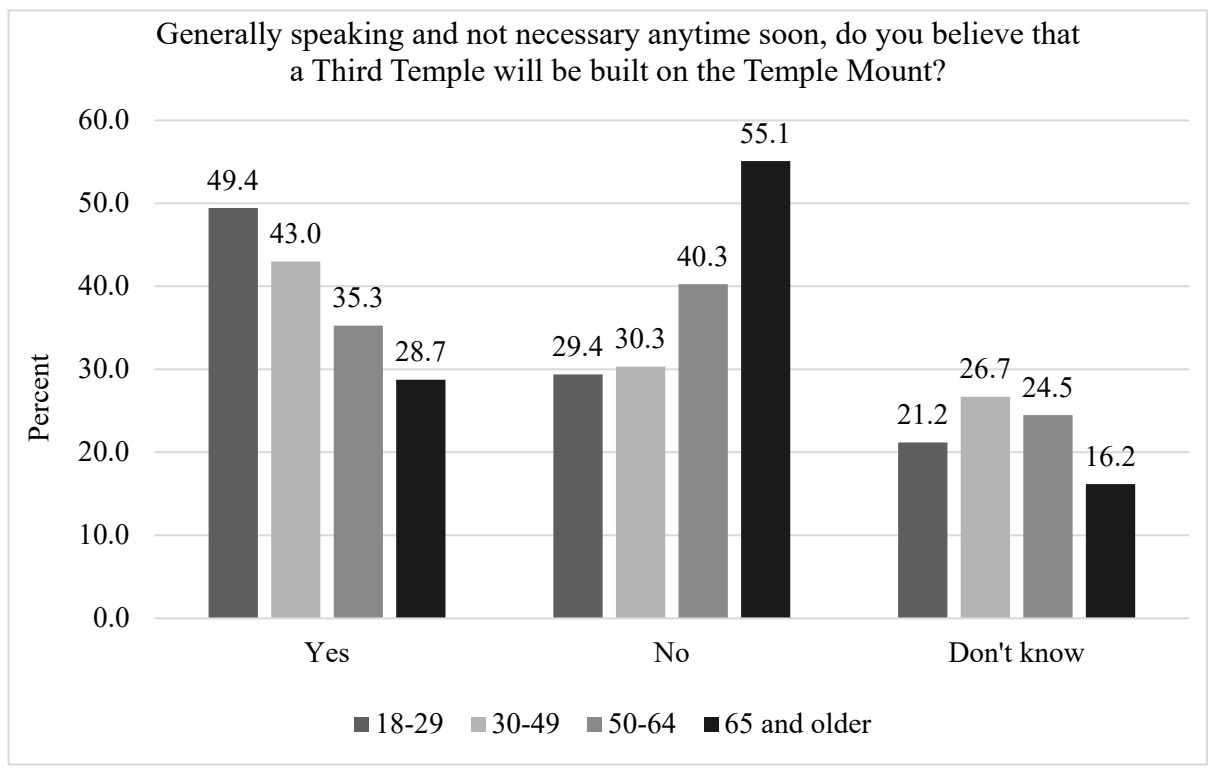

**Figure 11.** Age and Third Temple.

It is interesting to see the breakdown of the party vote and the belief in the Third Temple. Table 13 brings again the religious–secular binary, unlike the question of visiting the Temple Mount discussed earlier.

**Table 13.** Third Temple and Party Vote.

|  | Shas | Yahadut Hatorah | Hatzyonut Hadatit | Yisrael Beitenu | Likud | Yesh Atid | Hamachane Hamamlachti | Haavoda | Metetz | Hadash |
|---|---|---|---|---|---|---|---|---|---|---|
| yes | 85.1% | 89.3% | 67.4% | 19.4% | 49.6% | 11.9% | 18.1% | 6.1% | 6.5% | 0.0% |
| No | 2.7% | 2.9% | 9.7% | 64.5% | 18.2% | 70.3% | 59.8% | 57.1% | 83.9% | 0.0% |
| I don't know | 12.2% | 7.7% | 22.9% | 16.1% | 32.3% | 17.8% | 22.1% | 36.8% | 9.7% | 100% |

A follow-up question was given only to those who agreed with the Third Temple statement. The traditional belief is that, at the End of Time, the messiah would build the Third Temple with an act of miracle. This passive position does not require human action to promote the messianic timeline. Since the State of Israel took the Temple Mount complex in the Six-Day War, most Orthodox authorities (which included ultra-Orthodox authorities side by side with religious–Zionist authorities, including Rabbi Tzi Yehuda Kook, the spiritual father of the Gush Emunim movement) agreed with the passive approach. Gush Emunim was conflicted. On the one hand, it saw the Israeli victory as a sign from God for the upcoming redemption. On the other hand, on the question of Temple Mount, it adopted a passive approach (Inbari 2009, pp. 17–30). However, since the 1980s, a small circle of activists has pushed toward holding a more active role in the preparations for the Third Temple. Most of these activists come from religious–Zionist circles, and the most visible manifestation of their work is with the Temple Institute, a museum and an institution that prepares Temple vessels and trains priests for their rituals once the Temple is ready while finding and growing the red heifer. These activists demand that the State of Israel take charge and start preparation for the Third Temple as a national duty (Chen 2017).

We followed up with an additional question among those who agreed with the idea of a Third Temple: Who will build the Temple?

The results in Table 14 show that most of the respondents who agree with the traditional belief that, in the unseen future, a Third Temple would be standing on the Temple Mount, believe in the passive traditional expectation that this is the role of the messiah to build it. About a quarter of the respondents agree with the more proactive approach that puts the building enterprise onto the shoulders of the State of Israel. These respondents represent about 11% of the total sample.

**Table 14.** Who will build the Temple?

|  | **By the Messiah** | **By Israel, before the Coming of the Messiah** |
|---|---|---|
| How, in your opinion, will the Temple be built, is it: | 72.7% | 27.3% |

## 7. Discussion

This study shows that, among the Israeli public, there is a so-called secular–religious dichotomy, at least to some extent. We were able to confirm that about 50% of the sample prays often, believes that Judaism is the only true religion, and identifies as traditional or Orthodox. This bloc also agreed with most of the traditional statements regarding personal salvation and general beliefs in the coming of the messiah.

However, those who identify as secular are more complicated to classify, and secularity can mean many things in the Israeli context. Rosner and Fuchs say that this group ought to be divided into two sub-categories—total secular and somewhat traditional secular. Most seculars, say Rosner and Fuchs, are not anti-tradition. These two groups oppose religious legislation and do not see the Halakhah (Jewish religious law) as binding, so it is common to see them travel on Shabbat. However, on observance of rituals, like celebrating

religious holidays, there is a difference between the two groups where the totally secular observe fewer holidays, Kashrut, or recite the Kadish for a dead parent. Rosner and Fuchs emphasize that secular Israelis are in conflict with the religious establishment in Israel, but they are in harmony with their Judaism when Judaism means celebrating holidays and key annual rituals (Rosner and Fuchs 2018, pp. 119–32).

In our study, the only ritual we asked about was prayer. Although many who identify as secular never pray or pray seldomly, they may still hold supportive views on religious matters. Rosner and Fuchs said that secular Israelis believe in God less than religious or traditionalist Jews, but many of them *are* believers, or at least, are not *non*-believers (Rosner and Fuchs 2018, p. 119). Our study shows that, indeed, most secular Jews agree with statements about the belief in a God or higher power. Hagar Lahav called them 'secular-believers.' In that regard, the Israeli case is not so different from that of the U.S., where, according to the Pew Research Center, 90% of all Americans believe in the Biblical God or some higher power (O'Reilly 2018). In our study, 86% of Israelis embrace these beliefs. Many seculars also agree with the biblical ideas that identify Jews as "chosen people" who are living in the "Promised Land".

We found that it is useful to divide the secular label into three groups: traditionalist seculars, spiritual seculars, and atheist or agnostic seculars. Traditionalist seculars are those who agree with the tradition while holding a non-observant way of life. Spiritual seculars identify with faith but are more critical of traditional positions, and agnostic or atheist seculars are the most critical of traditional Jewish positions. Therefore, one can agree on some traditional positions, like the belief in the biblical God, while being skeptical toward the afterlife, as an agnostic would. The type of secularity of Israeli Jews can change on subjects. There is a spectrum of opinions, from support for traditional views to complete atheism, and people can move between the sub-categories. Our study confirms the "post-secular" theories of hybrid secular identities.

When it comes to the eschatological statements in the survey, in most cases, the seculars predictably disagreed with them. Their opinion was divided between opposition or "I don't know". Rosner and Fuchs observed that most Israeli seculars feel comfortable with celebrations of Jewish holidays, which, in Israel, are practiced as national holidays, and with private observance of some rituals. We can thus conclude that Israeli seculars feel more comfortable with the general notions of Judaism, such as God and Election, but feel less comfortable with traditional, maybe even "superstitious", beliefs in the unseen.

We can only speculate as to why, when it comes to eschatology, Israeli seculars are opting for atheist and agnostic views. It is logical and not surprising that the issues of End Times bear little relevance to them. An atheist denies the existence of God, while an agnostic would argue that there is no proof for the existence of God. Those who rejected eschatological beliefs might have acted as atheists, while those who chose the "I don't know" option might have been more skeptical, as agnostics would.

The essence of Judaism among seculars does not require accepting all the premises of tradition. Thus, popular eschatological beliefs might not be as articulated as Orthodox followers would. Also, as this study has shown, age and income factors need to be accounted for. It is possible that going up the socioeconomic ladder also entails a more skeptical point of view toward popular religious beliefs, which might be counted as something that belongs to the lower classes. The hostility of the seculars toward the religious establishment might play a subconscious role as well in rejecting popular religion, as eschatological beliefs might be viewed as part of the "Orthodox package" and, thus, as something that ought to be opposed.

This study also showed that there is an age factor to account for when it comes to questions of faith. The youngest Israelis comprise the most religious age cohort, and the older generations are the least religious. This phenomenon might be the result of the fast population growth of Orthodox and traditional circles and the consequence of almost fifty years of right-wing coalitions and control of the educational system in Israel by Orthodox

parties since 1977. Does this mean that Israel's Judaism will become more Orthodox in a generation or two? Possibly, but not necessarily. Only time will tell.

The question of the Temple Mount breaks the binary views and mixes religious sentiments (for or against visiting the site) with other opinions. The traditional view is that Jews should not visit the site due to issues of ritual purification. We saw that most ultra-Orthodox Yehadut Hatorah's voters objected to visiting the site, probably because of the traditional views. The site is also a focus of tension and friction in the Israeli–Palestinian conflict. Those coming from a left-leaning position might not wish to provoke the Palestinians (possibly Meretz voters, who also overwhelmingly oppose visiting the site). Half of the secular and even anti-clerical Yisrael Beitenu voters supported visiting and praying on the Mount, as well as 60% of the Likud voters. Our data do not directly explain why such support exists, but a way to explain this is by attributing it to a matter of national pride. Allowing de-facto Muslim control over the site, which is the holiest site in the Jewish religion, while not allowing Jews to pray on it, might have brought these supporters to agree with praying on the holy site as an act of showing Jewish ownership.

In the survey, we asked two questions to examine the support for ideas that the State of Israel should initiate the construction of a Jewish structure on the Mount—whether a synagogue or a Temple. We found that 10–20% of the sample supported such ideas, with the most significant support coming toward the idea of a synagogue on the site.

Since the 2000s, there has been a more robust engagement toward the idea of Jewish presence on the Temple Mount, when young religious–Zionist activists visit the site and engage in solitary prayer. On Jerusalem Day 2022, a new record was broken when 2626 men and women, mostly religious–Zionist youth and young adults, entered the Temple Mount, while about 1000 more eventually were not allowed in after the allotted visiting time passed (Segal 2022). Since the Mount was reopened for Jewish visits in 2003, the attendance records have been broken every year, and, in 2023, the record rose to 50,000 visits, according to the activists (a number we cannot verify independently). Although these are small numbers in comparison to the 11 million visitors to the Western Wall in 2023, this is clearly a growing movement. Israeli Knesset members often visit the site, and some government ministers have visited the site in recent years.

This survey was conducted six months before the tragic events of 7 October 2023, and the outbreak of the Israel–Hamas war. It is still early to speculate how these traumatic events, where 1200 Israeli civilians were brutally murdered, raped, and abused, while 240 were kidnaped, will influence Israeli politics and piety. The trauma of the 1973 Yom Kippur War has created significant political and spiritual changes in Israel's landscape. Scholar Benjamin Beit-Hallahmi found that the trauma created a shift toward greater religiosity, with a large *tshuva* movement of seculars discovering their religion and joining the Orthodox circles, hand-in-hand with a large expansion of New Age and Asian spirituality like Krishna and Buddhism (Beit-Hallahmi 1992). Early signs are that a similar response is developing in Israel as a result of the ongoing Israel–Hamas war in Gaza. A poll conducted by the *Jerusalem Post* shows a rise in religiosity since 7 October (more specifically, a stronger belief in God) among one-third of the polled Israelis (Klein 2024). Only further polling, over a longer period of time, would allow us to examine the extent to which the trauma of this horrific event has changed Israeli–Jewish perceptions and in what direction.

**Author Contributions:** All authors contributed equally to this work. All authors have read and agreed to the published version of the manuscript.

**Funding:** All Israel News and Chosen People Ministries.

**Institutional Review Board Statement:** The study was conducted in accordance with the Declaration of Helsinki, and approved by the Institutional Review Board of The University of North Carolina at Pembroke protocol code FWA 00005281, 16 March 2023.

**Informed Consent Statement:** Informed consent was obtained from all subjects involved in the study.

**Data Availability Statement:** The original contributions presented in the study are included in the article, further inquiries can be directed to the corresponding author.

**Conflicts of Interest:** The authors declare no conflicts of interest.

## Notes

1    GeoCartography is a member of ESOMAR and the Israeli Research Institutes Association. All Israel News and Chosen People Ministries sponsored the survey.

2    We considered whether income hides the effects of education but did not find this to be the case.

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
