# Peer review of "Israeli Jewish Attitudes toward Core Religious Beliefs in God, the Election of Israel, Eschatology, and the Temple Mount—Statistical Analysis"

_religions, doi:10.3390/rel15091076_

Round 1

Reviewer 1 Report

Comments and Suggestions for Authors

1) Perhaps authors would consider a more extended conclusion/implications for the future post-October 7 when the Hamas-Israel War is over.

2) Some possible more comments regarding the shutafut/partnership between the US and Israeli Jewish communities.

Author Response

Reader 1 made two comments as recommendations but not requirements on a more speculative matter: The first comment was to extend the conclusions for the future post-October 7, when the Israel-Hamas war will be over. We prefer not to engage in future speculations. When the war is over, we might survey again to see how the event impacts Israeli piety and in what direction. The second comment was to discuss how these findings might affect American and Israeli Jewish relationships in the future. We are now working on a large paper just on that question, and we ask the reader for some patience. This comment will be addressed in future publications.

Reviewer 2 Report

Comments and Suggestions for Authors

This is a very good empirical study.  I commend you for having the final paragraphs in the discussion regarding October 7th.  Given your study being conducted beforehand, it would have been easy to just offer a caveat saying the paper reflects attitudes before, but you attempted to gauge how attitudes might have changed in the wake of those horrors.  It offers a glimpse into what could be a rich followup to this study.

Author Response

Thank you for your comments and support in this research

Reviewer 3 Report

Comments and Suggestions for Authors

Israeli Jewish Attitudes toward Core Religious Beliefs in God, The Election of Israel, Eschatology, and the Temple Mount –Statistical Analysis

This paper addresses religious and political aspects with the aim of identifying deep-rooted processes currently taking place in Jewish society in Israel. At the core of the study is the presentation of findings from a large-scale survey based on a representative sample of 1,204 respondents. The article serves as a significant source of up-to-date information regarding the ongoing situation in a country central to one of the most challenging and persistent international conflicts. Therefore, it is of great importance and is highly recommended for publication, subject to some relatively minor revisions.

The most significant findings of the survey presented in the article, in my opinion, are those indicating that the country is undergoing a pronounced process of religious intensification. In this process, younger cohorts express higher levels of religiosity than older Israelis (line 320). The article does not discuss whether this phenomenon is a result of faster natural population growth among religious and right-wing groups, or if it is the consequence of over a quarter-century of governance by coalitions of right-wing and religious parties in Israel—a point that warrants at least some reflection and brief discussion.

At a current level, other important findings include attitudes toward the Temple Mount and the prayers of Jews on the site. At a time when one of the most secular parties in the country (Yesh Atid), in collaboration with an ultra-Orthodox party (Shas), seeks to turn a religious halakhic prohibition against Jewish prayer on the Temple Mount into civil law, the survey’s findings are especially significant. They reveal that as the age of respondents decreases, so does opposition to such prayers, as well as to other measures reflecting Israeli control over a site sacred to Muslims. If, as the data suggests, there is a growing willingness among Israeli Jews to infringe upon Muslim autonomy on the Temple Mount, this could predict further escalation in the Israeli-Arab conflict.

Additionally, the distinction between different types of secular Jews -- traditional secular, spiritual secular, and atheist/agnostic secular (lines 576-580) -- is significant, largely new, and interesting. Given that the author appears well-versed in Hebrew, as suggested by the bibliography, I recommend that she consider Yuval Jobani's paper, Jewish secularisms: Philosophical and educational perspectives in Katz, Ratzabi, and Yadgar (2014), Beyond Halacha, and Shlomi Sasson’s (2019) The New Secularism.

The article is written in a clear and structured manner, but I think that the organization of the material does not do it justice. Firstly, both the abstract and the article itself open with well-known facts, particularly the almost equal division between secular and religious groups in Israel. The article solely dedicates its ending paragraph to the events of October 7, 2023, and the war between Israel, Hamas, Hezbollah, and Iran. I think this is a blunder. Although the survey was conducted before this date, many of its findings deeply resonate with the current situation in Israel. I believe that this connection should be made at the beginning of the article, not at the end.

Below are a few specific points, in addition to the general remarks:

·         The theoretical background can benefit from addressing theories about Israel in Yaakov Yadgar's (2020) book, Israel's Jewish Identity Crisis: State and Politics in the Middle East.

·         Lines 26-37: The claim that the beliefs in the afterlife, heaven and hell, and the coming of the messiah, are also essential Jewish ideas may be phrased more cautiously. While these beliefs do exist in Judaism, they are not universal or consistent across all contexts. In general, the author tends to take statements about beliefs literally, while since Maimonides, such statements have often been understood metaphorically rather than literally. Furthermore, this paragraph is in need of a sentence or two about post-Holocaust Jewish theology, which provides a distinct perspective on Jewish religiosity.

·         The author interprets the information presented by Rosner and Fuchs as a claim about secularization in Israel. This is not how I read them. From what I understand, they argue that Israel's Jewish community is developing a form of Judaism that incorporates religious, secular, and traditional aspects.

·         According to the table presented, there has been a significant increase in the number of those identifying as completely secular between 2017 and 2023. The author's own arguments and other surveys conducted in Israel in recent years contradict this figure. This discrepancy could be caused by methodological differences, but regardless, it necessitates attention and explanation.

·         Line 305 -- Table 1 and all subsequent tables: The article does not address the significance of the correlations it identifies, which should be rectified.

·         Line 313: The author writes that Apparently, there is a correlation between income and beliefs: the less prosperous Israelis are more religious, and vice-versa. As can be seen on the following page, this is not true for those who believe in a higher power that is not God. On the contrary, they tend to belong to higher social classes. This point requires further explanation and discussion.

·         In the same context, the author acknowledges that she has examined and eliminated the possibility that education plays a role in the relationship between belief and economic status. Has age been taken into consideration? In other words, can it be argued that given that younger people tend to be less financially established than older individuals, and according to the study’s findings, are more likely to believe in God, age might be the factor driving the correlation between belief and economic status?

In summary, it's an intriguing and usable paper that could be improved with a little work.

Be'hatzlacha

Author Response

Thank you for reading the manuscript and supporting it.

Upon your recommendation, we discussed in the conclusions why young Israelis tend to be more religious.

We added references to Yuval Jobani’s work (two items), and to Yaacov Yadgar’s work.

We addressed that Maimonides viewed the afterlife in an allegorical way.

We addressed the fact that people of higher income prefer to choose higher power over the biblical God.

Regarding the comment about the possibility of a correlation between education, income, and age, we run the statistical model again, and the Pearson R2 statistic does not show a multicollinearity problem or significant levels of correlation between income, education, and age.

Reviewer 4 Report

Comments and Suggestions for Authors

The paper offers a detailed and nuanced examination of the interplay between religion and secularism in Israel. Following are three minor suggestions that could enhance the manuscript, which already seems complete:

First, In the introduction of the paper, I suggest including what Michael Walzer called “The Paradox of Liberation”. Walzer argues that secular liberation movements often result in unexpected religious revivals. He explores how revolutionary leaders in countries like India, Israel, and Algeria underestimated the profound cultural influence of religion, which reasserted itself after their secular successes, creating a paradox where secular revolutions unintentionally provoke religious counterrevolutions, challenging the movements' original goals. See Michael Walzer, The Paradox of Liberation: Secular Revolutions and Religious Counterrevolutions. Yale University Press, 2015.

Second, in the framework of the discussion of “secular believers” (line 150) I suggest adding a reference to Yuval Jobani, The First Jewish Environmentalist: The Green Philosophy of AD Gordon. Oxford University Press, 2024, pp. 107-115 which deals with A.D. Gordon’s religious secularism.

Third, in lines 38-39, the term "otherworldly" would be more appropriate than "this worldly."

Author Response

Thank you for reading the paper and making revision suggestions.  

We added a reference to Michael Walzer and Yuval Jobani’s work upon your request.